

# A quantum theory of the nearly frozen charge glass

Simone Fratini[1⋆], Katherine Driscoll[1], Sergio Ciuchi[2] and Arnaud Ralko[1]

**1** Institut Néel, CNRS & Univ. Grenoble Alpes, 38042 Grenoble, France
**2** Dipartimento di Scienze Fisiche e Chimiche, Università dell'Aquila, via Vetoio,
I-67010 Coppito-L'Aquila, Italy and Istituto dei Sistemi Complessi,
CNR, via dei Taurini 19, I-00185 Rome, Italy

⋆ simone.fratini@neel.cnrs.fr

## Abstract

We study long-range interacting electrons on the triangular lattice using mixed quantum/classical simulations going beyond the usual classical descriptions of the lattice Coulomb fluid. Our results in the strong interaction limit indicate that the proliferation of quantum defects governs the low-temperature dynamics of this strongly frustrated system. The present theoretical findings explain the phenomenology observed in the $\theta$-ET$_2$X materials as they fall out of equilibrium, including glassiness, resistive switching and a strong sensitivity to the electronic structure anisotropy. The method devised here can be easily generalized to address other systems and devices where itinerant and correlation-localized degrees of freedom are intertwined on short lengthscales.

# 1 Introduction

The discovery of Mott insulating and Wigner crystal phases in Moiré superlattices [1–3] has brought it to the public attention that interacting electrons behave differently when they live in frustrated lattices, adding an extra layer of complexity to the already rich field of correlated quantum matter. The triangular geometry, that is found in these systems as well as in many other correlated materials, is known to be detrimental to spin ordering, and it shouldn't come as a surprise that the charge itself is also subject to analogous frustration effects: depending on filling factors and commensurability, the charge ordering patterns dictated by the strong Coulomb interactions present in these systems are frustrated, as they can either cooperate or compete with the underlying lattice periodicity. More generally and regardless of the particular lattice geometry, the interplay between different lengthscales — the natural lengthscale of electronic ordering doesn't necessarily match that of the underlying atomic or molecular lattice — can lead to metastable electronic phases that are intermediate between liquid and crystalline [4], one notable experimental example being the "hidden phase" found in the layered dichalcogenide 1T-TaS$_2$ [5]. From the theory standpoint, describing and understanding these systems where itinerant and correlation-localized electrons are intertwined on different lengthscales is challenging, as the powerful numerical techniques developed for strongly correlated systems are mostly tailored for local interaction effects.

In this regard, the organic metals of the $\theta$-ET$_2$X family host a puzzling charge glass state [6, 7] that still lacks a proper microscopic description. These materials are composed of layers of BEDT-TTF (ET) organic molecules whose ordered arrangement approaches a triangular lattice, with slight deviations that can be tuned by an accurate choice of the cation X. This is illustrated on the left panel of Fig. 1. At the concentration of one hole (three electrons) per two molecules determined by stoichiometry, the triangular geometry is known to frustrate possible electronic orders that would result from the strong Coulomb repulsion between the electrons [8]. The observed charge glass is particular in that it emerges in compounds that are apparently devoid of structural disorder: this implies the existence of some form of self-generated randomness, possibly originating from the dynamical arrangement of the electrons themselves. It has been proposed that the randomness underlying the glassy behavior could arise from the competition between the many metastable states that emerge when long-range order is frustrated [7,9–11].

Experimentally, the extensive role of metastable configurations in the $\theta$-ET$_2$X materials was suggested by the observation of a progressive freezing of electronic dynamics upon lowering the temperature, accompanied by aging phenomena, as well as the presence of diffuse signals in the X-ray spectra indicative of a continuum of competing (short-range) ordered structures [7]. Many experiments have explored in particular the dynamics of the approach to glassiness by making use of thermal quenches in order to avoid electronic crystallization [12, 13]. It is now understood that the velocity of the quench required to access the charge glass varies from material to material, and correlates to the degree of geometrical frustration set by the molecular lattice. The transport properties are also affected by the amount of frustration, and switching between resistivity values differing by several orders of magnitude can be observed when the system transitions from an electronically ordered to a metastable glassy state [7,14].

The underlying idea that charge frustration could be at the origin of the observed behavior was given a microscopic basis soon after the discovery [11], demonstrating the crucial role of *long-range* interactions between the holes. It was shown that the combination of long-range Coulomb interactions and the triangular geometry of the molecular lattice at one quarter filling leads to the emergence of very many competing metastable states with amorphous "stripe-glass" spatial structures and coexisting short- and long-range order. The dynamics of the resulting frustrated phase were explored through classical Monte Carlo (MC) simulations [11], which showed remarkably slow viscous dynamics typical of strong glass for-

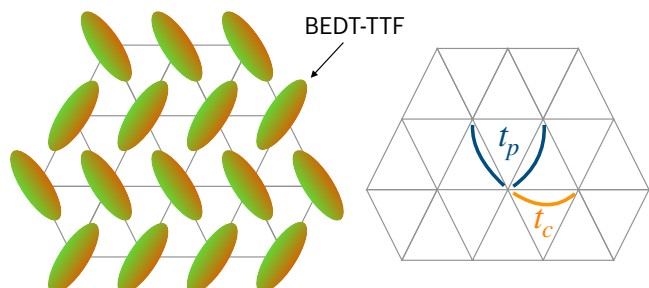

Figure 1: **Sketch of the system geometry.** In the compounds $\theta$-ET$_2$X family, the molecules (ET = BEDT-TTF) form a nearly triangular lattice (left) with intermolecular hopping integrals $t_p$ and $t_c$, respectively in the diagonal and the horizontal directions (right). The triangular geometry is also found in transition metal dichalcogenides and more generally in Moiré superlattice systems.

mers as well as aging phenomena characteristic of supercooled liquids on shorter timescales. This phenomenology, which strikingly resembles the experimental observations, was shown instead to be totally absent for short-range interactions.

If one thing is puzzling about the agreement found between theory and experiment is that it was based on purely classical simulations. The bandwidths in the $\theta$-ET$_2$X class range from few tenths to half an eV [15], and it can be argued that the quantum fluctuations associated with fast inter-molecular electron transfer should be able to melt the glassy state. Even more importantly, it is hard to explain on the basis of purely classical considerations the extreme sensitivity of the glass-forming ability to rather modest modifications of the structure [13,14]. The largest changes in the lattice anisotropy that can be obtained by chemical or physical pressure entail anisotropies in the intermolecular nearest-neighbor repulsion that do not exceed 15%, while the dynamical timescales at play, viz. the speed of quenches needed to drive the system out of equilibrium, vary by orders of magnitude from compound to compound.

Here we show that quantum fluctuations do not immediately melt the glass, provided that intermolecular transfer integrals are small compared to the dominant Coulomb interaction energy: this corresponds precisely to the strongly interacting regime that applies to the $\theta$-ET$_2$X organic metals. Promoted by quantum fluctuations, quantum defects emerge and proliferate, governing the physics of the electronic system in a way that depends dramatically on the degree of frustration, being controlled by the anisotropy of the band structure. The present results explain the puzzling phenomenology observed in the compounds of the $\theta$-ET$_2$X class, where strong variations of the intermolecular transfer integrals can be achieved upon application of chemical or physical pressure.

## 2 Model and method

We study the following Hamiltonian:

$$H = \sum_{\langle ij \rangle} \left( t_{ij}\, c_i^\dagger c_j + h.c. \right) + \frac{1}{2} \sum_{i \neq j} V_{ij} \hat{n}_i \hat{n}_j \,, \tag{1}$$

where the first term describes the hopping of fermions on a lattice (brackets stand for nearest neighbors only), $\hat{n}_i = c_i^\dagger c_i$ is the local density operator and $V_{ij} = V/|R_{ij}|$ the non-local Coulomb potential between electrons on molecular sites $R_i$ and $R_j$. We consider a triangular lattice as appropriate for the $\theta$-ET$_2$X class, with a concentration of one hole per two sites and

transfer integrals $t_{ij} = t_p, t_c$ respectively in the diagonal and horizontal directions (sketch in Fig. 1), neglecting the spin degrees of freedom. This customary approximation is justified by the fact that the double site occupations required for spin exchange processes are suppressed at concentrations away from integer fillings. We consider values of $t_p/V \ll 1$ and vary independently the anisotropy ratio $\eta = t_c/t_p$ which, for the materials under study, ranges in the interval $-1 \lesssim \eta \lesssim 1$ [16]. The expression Eq. (1) has an explicit positive sign for $t_{ij}$ as appropriate for holes.

The thermal behavior of the model Eq. (1) was studied in Ref. [11] through classical MC simulations, highlighting a marked tendency to glassy behavior at low temperatures. Here we want to address the effects of quantum fluctuations when $t_{ij} \neq 0$. To do so, we perform a strong coupling perturbation expansion, treating the kinetic term in Eq. (1) to lowest order in $t_p/V$, following a methodology that was successfully applied to study the quantum melting of the charge ordered state in the presence of long-range Coulomb interactions both in one [17, 18] and two space dimensions [21, 22]. This method was shown to correctly capture the proliferation of defects involved in the quantum melting mechanism at $T = 0$, providing results in quantitative agreement with full exact diagonalization.

We generalize here this strong coupling expansion to finite temperatures, by coupling it to classical MC simulations to account for the many different charge configurations that are thermally accessible beyond the classical ground state. We use local nearest-neighbor updates to mimic the hopping of electrons between molecules [23]. At each step in the MC evolution we then solve the one-body electron problem in the electrostatic potential determined by the collective distribution of classical occupations $\{n_i\} = 0, 1$, hence allowing the hole density to spread away from their classical positions on the molecular sites. This step is equivalent to the one that was used in the ordered case in Refs. [17, 18, 21] (where however only the configuration of minimal energy was considered), giving access to several physical observables such as the spatial distibution of the charge, the one-particle spectral function and the optical conductivity. Second, we include in the MC engine the kinetic energy gain enabled by the quantum spreading of the charge. This is achieved by replacing the classical statistical weight $\exp[-\beta E_{\{n_i\}}]$ in the Metropolis-Hastings sampling by the quantum corrected weight $\exp[-\beta(E_{\{n_i\}} + \langle K \rangle_{\{n_i\}})]$, where $\beta = 1/k_B T$, $E_{\{n_i\}}$ is the energy of the classical configuration (evaluated via standard Ewald summations on finite size lattices) and $\langle K \rangle_{\{n_i\}}$ is the thermalized quantum expectation value of the kinetic term in that same configuration. Physically, this means that at each step we work within the restricted statistical ensemble associated to a given metastable minimum in the classical configuration space, letting the electrons fully thermalize within this minimum.

The method devised here goes beyond the strong coupling perturbation approach used in [19,20]. It can be seen as a practical implementation of early ideas of Andreev and Kosevich [9] and Efros [10], who hypothesized a clear separation of time-scales between fast individual and slow collective motions, the latter being severely slowed down by the existence of many-body interactions at all distances. This "two fluid" behavior of the frustrated Coulomb system has been recently confirmed by a fully quantum study of long-range interacting electrons on the triangular lattice [22].

## 3 Results

### 3.1 Dynamical slowing down

One of the key features found in the classical Coulomb gas is the dynamical slowing down occurring upon cooling near the freezing transition at $T_c = 0.038V$ [11]. To assess if this

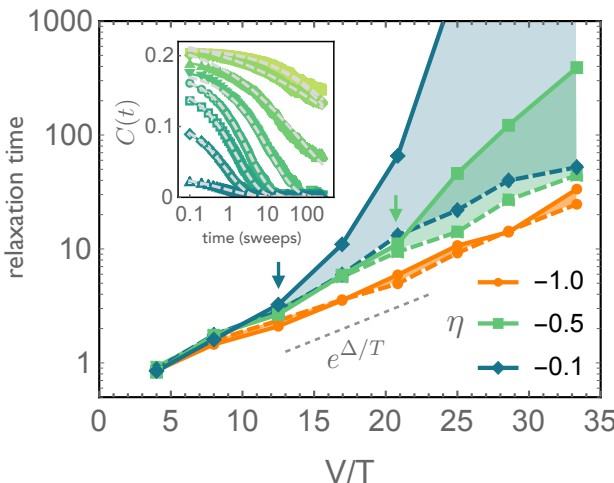

Figure 2: **Dynamical slowing down**. Arrhenius plots of the relaxation times as determined from stretched exponential fits of the local density-density correlation function for different degrees of anisotropy $\eta$. An example is shown in the inset ($t_p = 0.12V$, $\eta = -0.1$ and $t_W = 200$, all data evaluated over 250 sweeps and averaged over $\sim 100$ independent walkers; temperatures decreasing from bottom to top, fits shown as dashed lines). Dashed and full lines correspond respectively to waiting times $T_W = 20, 200$. All simulations are performed on a regular $L \times L$ cluster with $L = 12$. Time is measured in sweeps, where one sweep $= L^2$ update attempts.

phenomenon is preserved in the presence of quantum fluctuations we evaluate here the local autocorrelation function $C(t, t_W) = \sum_i \langle n_i(t + t_W) n_i(t_W) \rangle / N$, where $t_W$ is the waiting time the system is allowed to relax after initializing it to a random initial configuration [11,23].

Panel Fig. 2 reports the relaxation times obtained for different values of the anisotropy ratio $\eta$ (labels and different colors) and different waiting times $t_W = 20, 200$ (dashed and solid respectively), as extracted via stretched exponential fits of the form $C(t, t_W) = C_0 \exp(-(t/\tau)^\alpha)$ with $\alpha < 1$ (inset). For isotropic band structures ($t_p = 0.12V$, $\eta = -1$, orange), the relaxation time grows in an Arrhenius fashion upon cooling, typical of strong glass formers in the supercooled liquid regime. This behavior is analogous to the one obtained in the classical case, albeit with a considerably reduced energy barrier ($\Delta \simeq 0.11V$ instead of $\Delta \simeq 0.2V$) [11], implying much faster relaxation processes. Indeed, the exponential behavior is the same independently on the waiting time, indicating that the quantum equilibration occurs at times faster than $T_W = 20$ in the whole temperature window studied.

The situation changes drastically upon the introduction of electronic anisotropy (green and blue). Here a dynamical crossover temperature emerges below which the relaxation time strongly depends on $t_W$, indicating that the system falls out of equilibrium (arrows). Moreover, the relaxation time for long waiting times grows faster than exponential upon cooling, and the autocorrelation itself deviates from the stretched exponential form. The origin of this quantitative and qualitative dependence of dynamic relaxation on electronic anisotropy is addressed in the next paragraphs.

## 3.2 Emergence of quantum defects

Fig. 3(a) shows the evolution of the kinetic energy $\langle K \rangle$ as a function of simulation time in the case of an isotropic band structure. Starting the simulation from a perfectly ordered horizontal stripe configuration (gray), $\langle K \rangle$ shows discretized jumps from a constant baseline, signaling

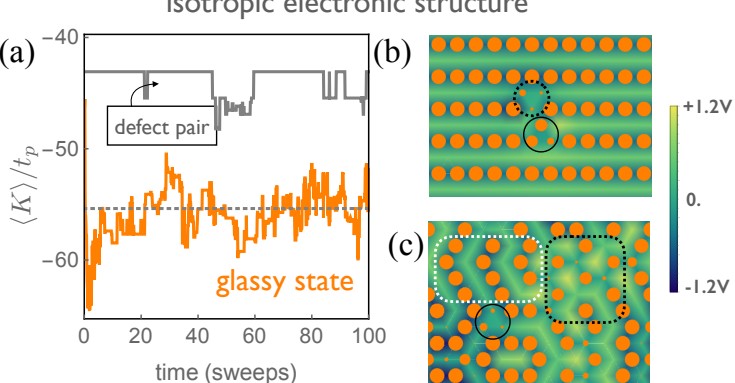

Figure 3: **Kinetic evolution and emergence of glassiness.** (a) Thermalized kinetic energy as a function of MC time for a single walker initially prepared in the perfect horizontal stripe arrangement (gray) or in a maximally disordered configuration (orange), for $t_p = 0.09V$, $\eta = -1$ and $T = 0.03V$, lower than the classical charge ordering temperature $T_c = 0.038V$. The gray dashed line is the long time average of $\langle K \rangle$. (b,c) Typical charge configurations in the two cases (radius of the orange disks proportional to the local hole density, the background map is the collective electrostatic potential).

the creation and annihilation of defect pairs with energy of order $t_p$. Their shape is shown in panel (b): Classically they correspond to moving a particle from a site on a charge rich stripe to a neighboring empty stripe. This creates an electrical dipole constituted of two oppositely charged defects (full and dashed circle; cf. [22], and [21] for the analogous defect pairs on the square lattice). Specific to the triangular lattice, this local fluctuation defines a region of three adjacent sites (full circle), all of them having three occupied neighbors and therefore an almost degenerate electrostatic potential $\phi_i = \sum_j V_{ij} n_j$ (these configurations are degenerate up to next nearest neighbor corrections). Quantum-mechanically, a metallic droplet can be formed through hybridization of the electron wavefunction on these three sites, gaining an energy $E_d \propto t_p$.

To mimic a rapid quench starting from high temperature, we also report in Fig. 3(a) the evolution of $\langle K \rangle$ starting from a random configuration (orange). After a sharp initial drop, the kinetic energy reaches a stationary regime exhibiting large fluctuations around an average value (dashed line) that is much lower than that of the ordered stripe configuration: Both features are indicative of the presence and dynamics of a large number of defects. The corresponding configuration map shown in panel (c) shows finite-size striped domains of random orientations coexisting with competing threefold order (respectively white and black dotted), in addition to several isolated defects (circle). It is important to note that quantum defects naturally arise at the crossings of stripes of three different orientations, which follows from the same neighbor counting argument presented above. The kinetic energy gain associated with threefold stripe crossings, and more generally with the nucleation and proliferation of extended quantum defects, is what makes the quantum stripe glass particularly stable in the maximally frustrated isotropic case: disordered configurations are favored as these enable an energy gain $\propto t_p$ that is larger than the $\propto t_p^2/V$ characteristic of ordered stripes.

Fig. 4(a) show traces analogous to the ones shown in Fig. 3(a), but now in a system with a highly anisotropic electronic structure. As in the isotropic case, the evolution starting from the ordered state (gray) shows successive processes of creation and annihilation of defects. More interesting is the evolution after a quench (orange): the kinetic energy here steadily decreases and reaches the same baseline value of the ordered state, indicating an unforeseen tendency

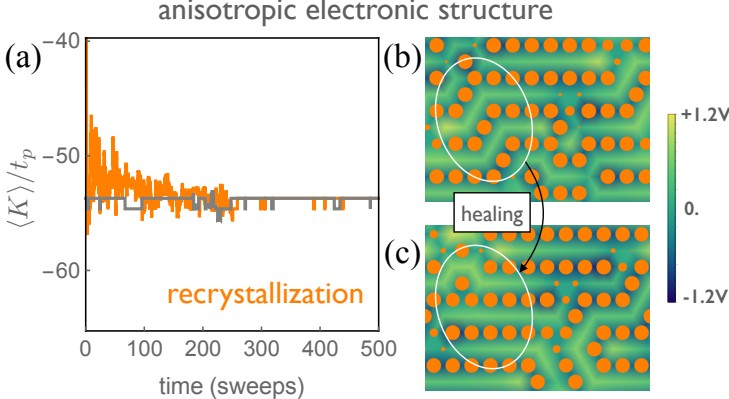

Figure 4: **Recrystallization by defects.** Same as Fig. 3, for an anisotropic band structure, $t_p = 0.12V$, $\eta = -0.1$. The two configurations shown correspond respectively to $t = 95$ and $t = 100$, illustrating the recrystallization through healing by anisotropic quantum defects.

to ordering. This healing process is illustrated in panels (b,c). Because of the anisotropy of the transfer integrals, the horizontal stripe direction is now favored over the other directions, since it maximizes the kinetic energy gain through (extensive) transverse fluctuations with probability $\propto (t_p/V)^2 > (t_c/V)^2$. We conclude that the same quantum defects that favored disordered stripe clusters in the isotropic case have an opposite effect when the electronic structure is anisotropic, being able to drive recrystallization into stripe order [12]. Along the process, charge fluctuations become less and less frequent and the relaxation time grows faster than exponential, as found in Fig.2.

## 3.3 Electronic properties and resistivity switching

To characterize the metastable electronic state found in the preceding paragraphs, Fig. 5(a) shows the distribution of local hole densities on the molecular sites, obtained after a quench in the isotropic case $\eta = -1$ ($t_p = 0.12V$, $T_W = 200$, $T/V$ labels as per panel (c)). At high temperatures (orange) the distribution is bell shaped and centered around the average hole concentration $n = 1/2$, indicating a normal fluid without charge order. Here the fluctuations are mostly of thermal origin, being entirely determined by the thermal fluctuations of the electrostatic potential, shown in Fig. 5(b). Upon cooling $P(n)$ progressively becomes bimodal due to local charge ordering, with sharp peaks at $n = \delta, 1 - \delta$, with $\delta \propto (t_p/V)^2$ as expected from second order perturbation theory. The distribution is however seen to saturate at very low temperatures due to the persistence of quantum defects, as the system never reaches the perfect stripe ordered state. This saturation occurs at a temperature $T \sim t_p$, signaled by a residual probability of finding local density values in the whole interval $0 < n < 1$, in agreement with recent experimental findings [24]. The saturation towards a glassy state is also seen in the distribution of electrostatic potentials, that remains disordered at all temperatures (Fig. 5(b)).

We now calculate how the incipient glassiness affects electron transport, making direct contact with available experiments. This is done in practice within the framework of transient localization theory [25–27]. The starting point consists in applying the Kubo-Greenwood formula to calculate the optical conductivity $\sigma(\omega)$ from the solution of the one-body electron problem in the electrostatic potential of the classical charges, which is obtained at each step of the MC evolution (see Sec. 2). Importantly, the Kubo-Greenwood treatment fully captures quantum localization corrections that are highly relevant for two-dimensional electrons subject

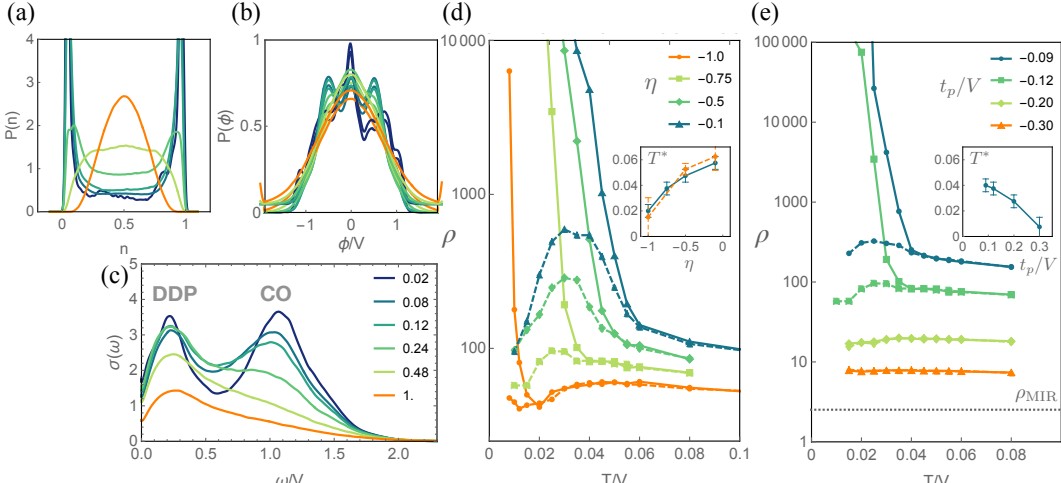

Figure 5: **Electronic properties and switching.** (a) Distribution of local densities (b) distribution of local electrostatic potentials and (c) optical conductivity spectra at different $T/V$ for $\eta = -1$ and $t_p/V = 0.12$ ($T/V$ values indicated by the labels on panel (c)). (d) Resistivity vs temperature obtained after a waiting time $T_W = 200$ starting from a quench (dashed) and an ordered initial state (full), for different $\eta$ for $t_p/V = 0.12$. (e) resistivity for $\eta = -0.75$ and different $t_p/V$. Units of $\rho$ are $c\hbar/e^2 \simeq 400\mu\Omega$cm, with $c \simeq 10$ the interlayer distance. The horizontal arrow marks the Mott-Ioffe-Regel limit, $\rho_{\text{MIR}} \simeq 1$ m$\Omega$cm. The insets show the dynamical temperature $T^*$ where the resistivity (blue) and the density correlations (orange dashed) become history dependent.

to a random potential [28]. In particular, these quantum corrections are responsible for the emergence of a localization-induced displaced Drude peak [26, 28] (see below), that would instead be missed by semi-classical approaches [19], and they have been suggested to be a likely cause of bad metal behavior [26].

The resistivity can be obtained in principle from the zero frequency limit of the optical conductivity as $\rho^{-1} = \sigma(\omega \to 0)$. This quantity, however, strictly vanishes within the present MC decoupling scheme because the random electrostatic potential extracted at each MC step is static. In reality the electrostatic environment is not static, as it varies on the scale of the collective rearrangements of the charge. The relaxation time approximation [25–27] provides an effective framework to account for the collective electron dynamics provided that these are much slower than the individual electron dynamics, which is precisely the realm of the present MC approach. In practice this is done by adding a Lorentzian broadening $\gamma$ to $\sigma(\omega)$, where $\gamma^{-1}$ represents the timescale of the collective charge motions that are at the origin of the random potential: the Lorentzian convolution mimics the effect of disorder dynamics on quantum localization corrections, allowing for a non-zero conductivity. We take the value $\gamma = 0.2t_p$ as determined from the fully quantum solution of the isotropic problem at $T = 0$ in [22], corresponding to a timescale $\gamma^{-1}$ that is consistently slower than that of the individual electrons.

Fig. 5(c) shows the optical conductivity $\sigma(\omega)$ calculated for the same microscopic parameters as in Fig. 5(a). The spectra exhibit two separate peaks of different nature: (i) A charge ordering (CO) peak at $\omega = V$, corresponding to the local fluctuation of holes in a short-range ordered, striped environment; this peak is washed out upon heating at $T \sim V$, i.e. when short-range order disappears and (ii) A displaced Drude peak (DDP) at $\omega = 2t_p$ that persists at all temperatures, probing the dynamics of holes trapped by the self-generated electrostatic

disorder $\phi_i = \sum_j V_{ij} n_j$, whose distribution is illustrated in Fig. 5(b). This is a microscopic realization of the disorder-induced DDP that is generally predicted to occur whenever electronic carriers interact with slowly fluctuating degrees of freedom [26, 29], embodied here by the slow collective fluctuations of the frustrated Coulomb fluid. The shape and T-dependence of the calculated DDP are strongly reminiscent of the one observed in the isotropic compound $\theta$-ET$_2$I$_3$ [30].

We are now in a position to illustrate the transport properties of the model, highlighting a strong dependence on the cooling dynamics as observed in experiments. Figs. 5(d,e) show the resistivity curves obtained from the procedure described above, (d) for varying levels of electronic anisotropy $\eta$ at $t_p = 0.12V$ and (e) for varying $t_p$ at fixed $\eta = -0.75$. For each choice of parameters we report results corresponding to a thermal quench (dashed), and those obtained instead by starting the simulation from the ordered state that minimizes the electrostatic energy (full), that are representative of experiments done at slow cooling rates.

In all cases, the resistivity curves show switching between a low resistance metastable state (quench) and a high resistance stable state (order) at a temperature $T^*$ that depends crucially on the microscopic parameters $\eta$ and $t_p$, cf. insets in panels (d,e). Isotropic electronic structures and large transfer integrals (i.e., large quantum fluctuations) both favor the emergence of quantum defects, leading to a reduction of $T^*$ and to an overall decrease of the absolute value of the resistivity. Almost T-independent resistivity curves with values close to the Mott-Ioffe-Regel limit are obtained in the most frustrated case, representataive of bad metal behavior. The features shown in Fig. 5(c,d) are in overall agreement with the experimental findings in the $\theta$-ET$_2$X series [14].

# 4 Conclusions and perspectives

The results presented here rationalize the extreme sensitivity of the electronic properties of the $\theta$-ET$_2$X salts to chemical strain, based on the proliferation and dynamics of quantum defects. This key aspect, that was unsatisfactorily assigned to the anisotropy of electrostatic interactions [13, 14], can instead be understood as an inherently quantum phenomenon.

More generally, the present theory highlights the key role played by quantum fluctuations in the competition between glassiness and charge order occurring in frustrated interacting electron systems. In addition to the importance of this result *per se*, the implications of the concepts and methods developed here could be quite far-reaching. First, the present work provides a reliable procedure to address bad metallic transport of quantum electrons in the presence of incipient glassiness, illustrated here on a system where such glassiness is caused entirely by their mutual long range interactions. Second, many physical systems exist displaying states where itinerant and correlation-localized electrons are intertwined on nano/mesoscopic lengthscales, with emergent short range order. In these systems, existing affordable techniques that are able to address the necessarily large system sizes, such as classical Monte Carlo on one side and fully quantum but weakly interacting methods on the other, are for opposite reasons unsuccesful. It will therefore be meaningful to apply the present method to other systems exhibiting a similar phenomenology, one example being the "hidden" metastable phase reported in 1T-TaS$_2$ close to Wigner crystal ordering [5], showing defect dynamics not captured at the classical level [31]. Similar ideas should also find relevance in Moiré superlattices near Wigner crystal phases [1–3], as well as in other quantum correlated devices with features on the few nanometer scale.

# Acknowledgements

The authors are grateful to K. Kanoda and D. Mihailovic for their stimulating input. K.D. acknowledges the European Union's Horizon 2020 Research and Innovation program under Marie Skłodowska-Curie Grant No. 754303.

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
