# Peer review of "A quantum theory of the nearly frozen charge glass"

_SciPost Physics, doi:SciPost Phys. 14, 124 (2023)_

## Round 1 · Referee Report · Anonymous (Referee 1) · 2022-11-3

Strengths

1- the subject is interesting 2- the results are sound and the physics is clearly explained

Weaknesses

1- the spin degrees of freedom are neglected without a discussion about possible drawbacks 2- the calculation of the optical conductivity and the assumption behind it are not discussed

Report

I found that this piece of work is timely, interesting and sound. The physical scenario seems to be explained in a thorough and comprehensible manner, although I feel that there is room for improving the clarity of the presentation. Overall, the criteria for publication in SciPost are met, so I would recommend publication provided the authors address the following issues:

1- Maybe readers not expert of specific physical systems but interested in the subject would benefit from some more detail (maybe even figures) to illustrate the nature and structure of, e.g., $\theta$-ET$_2$X and BEEDT-TTF (ET) organic molecules.

2- The Hamiltonian, Eq. (1), may be better discussed; although rather standard, I would explain the symbol $\langle ij\rangle$, I would put the first tw terms in parentheses, $\left(t_{ij}c^\dagger_i c_j+ h.c.\right)$, I would explicitly state that the term $i=j$ is excluded from the second sum, and I would anticipate the sketch in Fig. 3, as there are no severe space limitations, one might devote a figure to explain the toppings $t_c$ and $t_p$.

3- The authors state that they neglect the spin degrees of freedom without further comment, maybe the reader would benefit from a discussion about this assumptions and the possible drawbacks.

4- The authors say that in Fig. 2(a) $\langle K\rangle$ reaches a stationary regime, maybe a line showing the average value around which this quantity fluctuates might help the reader.

5- The authors give very little detail about their evaluation of the optical conductivity, it is not easy for the reader to make a definite idea about what is included and what is missing in their approach.

Requested changes

1- Provide more details about the physical systems to which the theory may apply, maybe adding explanatory figures.

2- Improve the discussion about the Hamiltonian, anticipating the sketch in Fig. 3 as an independent figure.

3- Discuss the assumption of neglecting the spin degrees of freedom and its possible drawbacks.

4- Improve the readability of Fig. 2, adding a line to highlight the average value of $\langle K\rangle$ in the stationary glassy state.

5- Improve the discussion about the calculation of the optical conductivity, in particular, the approximations involved in this calculation.

  • validity: high
  • significance: high
  • originality: high
  • clarity: good
  • formatting: good
  • grammar: excellent

Author:  Simone Fratini  on 2022-11-16  [id 3030]

(in reply to Report 1 on 2022-11-03)

We are glad that our work has been positively appreciated by both referees. It is our pleasure to submit a new manuscript with the requested changes proposed in the reports.

Report and requested changes:

1- We have added a new Fig. 1 where the molecular arrangement is provided together with a sketch of the microscopic hopping processes considered in the model. In the caption we have provided a brief list of physical systems to which the theory may apply.

2- The Section model and methods has been updated with more details about the symbols and explicit reference to the newly added Fig. 1.

3- We have added the sentence : "This customary approximation is justified by the fact that the double site occupations required for spin exchange processes are suppressed at concentrations away from integer fillings."

4- We thank the referee for this useful suggestion. The corresponding figure has been updated accordingly.

5- A new paragraph has been incorporated in Section 3.3 explaining the physical content and practical method of calculation of the optical conductivity and electrical resistivity, which is now also mentioned in the concluding remarks. Two new references have been added for the reader to find more details.

---

## Round 1 · Referee Report · Anonymous (Referee 2) · 2022-11-14

Strengths

1) To my knowledge, this work is the first to clearly address the quantum nature of charge glass with numerical methods. 2) The non-equilibrium nature of the inhomogeneous charge distribution is clearly demonstrated. 3) Remarkable agreement with experimental results.

Weaknesses

I do not find particular weakness.

Report

It is an issue of profound interest whether Coulomb interacting electrons on a regular lattice without disorder can form glasses. In recent years, several experimental studies indicated the emergence of such states in organic triangular-lattice compounds. Theoretically, glass formation of electrons on a triangular lattice is suggested in the classical limit (Ref. 11). An intriguing issue is whether such a glass state emerges with the transfer integrals included in the model as in real materials. In my opinion, the present work is giving groundbreaking results in the following respects.
First, the authors showed that even in the presence of finite transfer integrals (quantum fluctuations of charges), electrons can form a glass state and occasionally the quantum nature may even serve to stabilize the glass state. Second, through the numerical measurements of the relaxation rate, they found that the stability of the glass is quite sensitive to the anisotropy and magnitude of the transfer integrals, explaining experimental results. Third, the authors gave a conceptual interpretation to the numerical results, in the light of the generation and proliferation of quantum defects. Fourth, the calculated charge-density profile and resistivity qualitatively explain the experimental results of theta-ET2X both in their temperature dependence and anisotropy dependence.
I think that this work is making great contribution to the physics of charge frustration at large. The paper is well written and the referees are adequate. I recommend publication of this manuscript in this journal.

Requested changes

In Fig.4(a), the authors showed the temperature dependence of the distribution of charge density. The narrowing of the distribution at higher temperatures comes from the quantum effect, not from the thermal motional narrowing? Because this is an important point, I recommend the authors to give some explanation on this.

  • validity: high
  • significance: top
  • originality: top
  • clarity: high
  • formatting: excellent
  • grammar: excellent

Author:  Simone Fratini  on 2022-11-16  [id 3031]

(in reply to Report 2 on 2022-11-14)

We are glad that our work has been positively appreciated by both referees. It is our pleasure to submit a revised manuscript with the requested changes proposed in the reports.

Requested changes:

The width of P(n) observed at high temperatures is of thermal origin, being entirely determined by the distribution of local electrostatic potentials. For this reason, it is actually broader than the width of the sharp peaks observed at n=0.1 at lower temperatures (see Fig. 5(a)). To clarify this important point we have updated Fig.4 (now Fig.5) with a new panel (b) illustrating the distribution of the electrostatic potentials that is at the origin of the behavior of the charge density distribution. We have accordingly discussed the new figure in the manuscript.

---

## Round 2 · Referee Report · Anonymous (Referee 1) · 2022-12-2

Strengths

1- Relevance of the topic 2- Evidence in favor of the glassy nature of the charge frozen phase 3- Coherence and simplicity of the overall scenario

Weaknesses

1- Lack of a deeper discussion about the role of the geometry and/or the presence of other therms in the model Hamiltonian

Report

I am satisfied with the authors' reply to mi suggestions and criticism and with the changes they made. I understand that some of my curiosities go beyond the scope of the present manuscript, so I recommend publication without further changes.

---

## Round 2 · Author Response

Dear Editor,
We are glad that our work has been positively appreciated by both referees. It is our pleasure to submit a revised manuscript today with the requested changes proposed in the reports.

We have carefully addressed all questions and remarks and updated the manuscript accordingly. A complete list of changes is provided in our response to referees.

We are confident that all points have been treated and that this revised version of the manuscript is now suitable for publication in SciPost.

Best regards,
The authors.

---

## Round 2 · List of Changes

Report 1

1- We have added a new Fig. 1 where the molecular arrangement of the ET2X system is illustrated together with a sketch of the microscopic hopping processes considered in the Hamiltonian. In the caption we have given a brief list of physical systems to which the theory may apply.

2- The section Model and Methods has been updated with a corrected form of the Hamiltonian, including detailed explanations of the symbols and making explicit reference to the newly added Fig.1.

3- We have added the sentence : "This customary approximation is justified by the fact that the double site occupations required for spin exchange processes are suppressed at concentrations away from integer fillings."

4- We thank the referee for this useful suggestion. The corresponding figure has been updated accordingly.

5- A new paragraph has been incorporated in the manuscript in Section 3.3, providing full details on the physical content and practical method of calculation of the optical conductivity. The derivation of this original method is now also mentioned in the conclusion. Two new references have been added for the readers to find more details.

Report 2

The width of P(n) observed at high temperatures is of thermal origin, being entirely determined by the distribution of local electrostatic potentials. For this reason, it is actually broader than the width of the sharp peaks observed at n=0.1 at lower temperatures (see Fig. 5(a)). To clarify this important point we have updated Fig.4 (now Fig.5) with a new panel (b) illustrating the distribution of the electrostatic potentials that is at the origin of the behavior of the charge density distribution. We have accordingly discussed the new figure in the manuscript.

---

## Editorial Decision

published